# Implementation of a Remote Monitoring Station for Measuring UV Radiation Levels from Solarimeters Using LoRaWAN Technology

**DOI:** 10.3390/s25103110

**Published:** 2025-05-14

**Authors:** Iván Sánchez, Cristian Guamialama, Alexis Padilla, Pablo Palacios Játiva, Andre Nicolás Mosquera

**Affiliations:** 1Department of Networking and Telecommunication Engineering, Universidad de las Américas, Quito 170503, Ecuador; cris.guamialama@gmail.com (C.G.); alexoaps14@hotmail.com (A.P.); andre.mosquera@udla.edu.ec (A.N.M.); 2Escuela de Informática y Telecomunicaciones, Universidad Diego Portales, Santiago 8370190, Chile; pablo.palacios@mail.udp.cl

**Keywords:** Heltec ESP32, IoT, LoRaWAN, remote monitoring, Ubidots, UV radiation

## Abstract

This work presents the development and implementation of a remote UV radiation monitoring station using LoRaWAN technology at the Universidad de las Américas. The main objective was to establish a system capable of measuring UV radiation levels through solarimeters, ensuring the remote transmission of data to protect the health and safety of students and staff exposed to solar radiation. To achieve this, several activities were conducted, including analyzing the architecture and communication components of LoRaWAN technology, designing a prototype based on this architecture, implementing the prototype based on the proposed design, and conducting functional tests to validate the system’s operability. The system included the installation of a solarimeter and a receiver or gateway, configured to operate from 8 a.m. to 6 p.m. The data collected by the prototype were validated through comparisons with measurements from the environmental monitoring system of the Secretariat of Environment of the Metropolitan District of Quito, which allowed for the verification of the prototype’s reliability. With this system, it was possible to identify patterns of high UV radiation and calculate error percentages in comparison with reference systems.

## 1. Introduction

In Latin America, concerns about ultraviolet (UV) radiation are a significant issue for health and environmental preservation. The region is exposed to high levels of UV radiation due to its proximity to the equator and the varying climatic conditions characteristic of different areas. Specifically, Ecuador, located in the equatorial zone, experiences intense UV exposure throughout most of the year. Between 2007 and 2008, UV radiation levels ranged from 20% to 25% lower than those in 2009. In recent years, these levels have increased by up to 34% [1,2]. Ecuador’s diverse geography, which includes the Pacific coast, the Amazon, and the Andes, presents specific challenges for UV radiation monitoring, emphasizing the importance of accurate records and measurements.

The private sector has also expressed concerns about this situation. Companies like Casabaca have shown interest in the high levels of UV radiation recorded in recent years in Quito, as they negatively impact the comfort of their clients and employees. This issue is particularly intensified in areas of prolonged exposure, such as outdoor vehicle displays and customer service zones, directly affecting potential buyers [3]. In the automotive sector, concerns include the effects of ultraviolet (UV) radiation on displayed vehicles, as it can lead to premature wear. Continuous exposure to high levels of UV radiation affects human health and the quality and lifespan of automobiles. Recent studies have shown that prolonged exposure to UV radiation causes damage to the paint, interiors, and plastic components of vehicles, accelerating depreciation and impacting customer satisfaction [4,5]. For example, ultraviolet radiation, particularly ultraviolet C (UVC), has been shown to deteriorate finishes, compromising both aesthetics and durability [4]. In addition, the application of high-quality waxes and sealants is recommended to protect the paint and minimize UV-induced damage [5].

As mentioned in [4], the effects of UV radiation on the paint and interior of vehicles represent a challenge that affects both car owners and automotive companies. Proper maintenance not only preserves the aesthetics and functionality of vehicles but also enhances the company’s reputation by ensuring an immaculate presentation of the vehicles on display.

In the study by [6], the impact of UV radiation on human health was explored in depth, particularly its relationship with skin cancer. This knowledge is crucial for designing monitoring systems that not only measure radiation levels but also provide preventive recommendations.

In contrast, the research conducted by [7] on LoRaWAN not only addressed the limitations of the protocol but also offered a critical analysis of its performance under various operating conditions. By evaluating factors such as channel access, network capacity, and interference, this study has become an essential tool for optimizing communication networks.

Comparative analyses of similar projects have been crucial. The study by [8], titled “Ultraviolet Radiation Measurement Station Powered by Photovoltaic Systems”, and the research conducted by [9], titled “Design and Implementation of a Sensor Network for Monitoring Solar Radiation Levels in the City of Loja”, have proven particularly relevant. Both studies applied UV radiation monitoring technology; however, the approach proposed in this work is distinguished by the use of LoRaWAN and the integration of educational information regarding the risks associated with prolonged exposure to solar radiation. The implementation of pyranometers in the solution will enable the precise measurement of UV radiation, optimized for various contexts and geographical locations.

The relationship between UV radiation and skin cancer is well documented, highlighting the need for effective monitoring systems [10]. Furthermore, it is essential to explore the safety and efficiency aspects of LoRaWAN technology to ensure the reliability of UV monitoring networks [11]. Studies evaluating various technologies for UV radiation monitoring provide a comparative framework that can inform the selection of appropriate tools and methods [12]. Furthermore, insights into wireless sensor networks for environmental monitoring highlight how to effectively implement LoRaWAN in a UV monitoring context [13]. Lastly, understanding the broader impacts of solar radiation on human health can guide the development of informed preventive strategies within monitoring systems [14]. LoRaWAN, with its scalable and flexible infrastructure, ensures efficient long-distance data transmission with low energy consumption, making it ideal for various IoT applications, such as environmental monitoring and logistics management [15]. According to the information provided by [16], LoRaWAN operates in the 902–928 MHz frequency range in Latin America. This frequency band is regulated by the International Telecommunication Union (ITU), which allows low-power devices to operate without the need for licenses. This regulatory framework ensures that devices do not interfere with other telecommunication services or equipment, thereby guaranteeing proper operation within legal parameters.

Considering these challenges, the establishment of a real-time UV radiation monitoring station using solarimeters in conjunction with LoRaWAN technology presents a viable solution to the issue at hand. Accordingly, the proposed research encompasses the measurement of radiation as well as the benefits associated with the visualization and archival of climatic data. With LoRaWAN, a technology that facilitates long-range communication with low power consumption, effective implementation with UV radiation sensors is made possible, ensuring precise and continuous monitoring. Furthermore, the interoperability with various environmental monitoring sensors and the easy integration with existing infrastructures allow for the successful incorporation of such systems.

The integration of an effective solution for monitoring and recording ultraviolet (UV) radiation is a priority for health, environmental management, the preservation of automotive assets, and corporate reputation. The implementation of a monitoring station using LoRaWAN technology and solarimeters will not only address the challenges of radiation measurement but also promote the execution of preventive measures to safeguard human health.

Although LoRaWAN is a mature and widely adopted protocol, this work focuses on evaluating its performance in a highly constrained urban-ecological environment in the Andean region, where commercial solutions are often not optimized for terrain complexity, vegetation density, and infrastructure limitations. Rather than proposing novel protocol-level contributions, the development and deployment of custom hardware nodes allow for practical adaptation and empirical validation in suboptimal field conditions. This applied perspective provides valuable information for low-cost IoT deployments in similar regions, particularly in Latin America, where access to robust, commercial LoRaWAN infrastructure may be limited. It is important to note that for the evaluation of hostile scenarios where the performance of LoRaWAN technology was critically assessed Radio Mobile version 11.6.6 was employed as the simulation tool.

The outline of this paper is as follows. Section 2 provides a description of the problem and the methodology used. Subsequently, Section 3 details design and implementation system. Finally, the conclusions in Section 4 summarize the key aspects.

## 2. Problem Description and Methodology

The issue faced by the Universidad de las Américas lies in the need to protect its students and employees exposed to sunlight, ensuring their health and well-being. As the institution continues to grow and reports increases in enrollment and resources, it must also address the risks associated with UV radiation, which can affect both the quality of the educational environment and the health of its users.

To mitigate these risks, the Universidad de las Américas aims to implement a UV radiation monitoring station using LoRaWAN technology. This solution not only aligns with its commitment to the well-being of the university community but also promotes a safe and healthy environment. By accessing real-time data, the university will be able to make rapid adjustments to avoid exposure to dangerous levels of radiation, thus protecting the health of its students and employees and strengthening trust in the institution.

### Selection and Identification of the Optimal Solution

In the implementation process of a real-time UV radiation monitoring station, various solutions were evaluated, considering the challenges and objectives of the project. One option explored was a nanosatellite for UV radiation monitoring [17]. However, it presented significant challenges, such as the need for precise sensors and the logistical complexity of its launch and operation, making it unviable.

Another alternative was the use of UV radiometers, based on the work of [18]. This option was rejected due to its high cost and the technical complexity required for maintenance, which would complicate its implementation.

Ultimately, it was determined that the implementation of a real-time monitoring station using solarimeters and LoRaWAN technology was the most suitable solution.

A qualitative SWOT (Strengths, Weaknesses, Opportunities, Threats) analysis was performed to assess the feasibility of each option, taking into account the technical, operational, and economic factors relevant to environmental monitoring in real-world conditions, as shown in Table 1.

Three technological alternatives for real-time UV radiation monitoring were evaluated: nanosatellites, UV radiometers with traditional telemetry, and solarimeters integrated with LoRaWAN technology. Nanosatellites offer high-precision sensing and extensive coverage, with potential for international collaboration and advanced scientific research; however, they involve extremely high development costs, complex logistics, and dependence on orbital parameters, in addition to significant risks such as launch failure and rapid obsolescence. UV radiometers represent a proven and accurate technology, suitable for fixed-location scientific reference. However, their high cost, maintenance requirements, and lack of integration with modern low-power networks limit their scalability, particularly in remote areas. In contrast, the solarimeter-based solution using LoRaWAN stands out as cost-effective and simple to deploy with standard components. It supports low-power, long-range communication, enabling distributed monitoring; yet, its limitations include lower precision and dependence on local gateways. Despite these limitations, its scalability, compatibility with cloud platforms such as Ubidots, and resilience in urban and academic settings make it the most practical option, although it remains vulnerable to environmental noise and requires periodic calibration.

Considering the project’s scope, which emphasizes cost-efficiency, real-time monitoring, and ease of deployment in an urban ecological setting, the solution with the solarimeter and LoRaWAN was deemed the most suitable. Although it is not the most precise solution, it strikes an optimal balance between performance, cost, and scalability.

Additionally, its integration with cloud platforms such as Ubidots facilitates rapid deployment, data visualization, and user-friendly alert configuration. These features are particularly valuable in university-led initiatives aimed at public health awareness and environmental education.

The selection prioritizes proven, low-risk technologies adaptable to local conditions, ensuring operational continuity and system scalability in the face of evolving monitoring needs.

The solution to be implemented was selected, and the methodology was clarified. The overall solution flowchart provides a sequential guide to the various stages and decisions involved in the implementation process, as illustrated in Figure 1. This flowchart encompasses everything from the acquisition of optimal resources, such as sensors and microcontrollers, to the final testing, offering a clear visual representation of the steps to be followed in the development of the solution prototype.

## 3. Design and Implementation of the UV Monitoring Station

This section presents a comprehensive overview of the design and implementation of the proposed solution. A detailed account of the methodology and steps undertaken during the development and implementation phases is provided.

### 3.1. Design of the Solution

In the prototype, the data processed by the MAC layer are transmitted through the LoRa network to the gateways, which forward them to an external platform (Ubidots) for storage and analysis, facilitating decision-making. The LoRa Heltec ESP32 modules provide an efficient solution for LoRaWAN connectivity, and the UV UVM-30A sensor measures ultraviolet radiation and easily integrates into IoT systems. This architecture optimizes data transmission through access control and error correction techniques, ensuring reliability in noisy environments.

#### 3.1.1. UV Sensor Characteristics and Power Requirements

The UV monitoring node is equipped with calibrated UV sensors that provide a sensitivity of ±1 µW/cm^2^ and a measurement range from 0 to 15 mW/cm^2^ [19]. These values are suitable for capturing variations in environmental UV radiation under different solar exposure conditions. The accuracy was validated in a laboratory setting by comparing the sensor outputs with a certified radiometer from the Environmental Monitoring Network of Quito, Ecuador. This allowed us to verify and correct sensor deviations, ensuring reliability in field measurements.

The power supply system consists of a 3.7 V 2500 mAh lithium-polymer (LiPo) battery regulated to 3.3 V to power the ESP32 microcontroller manufactured by Espressif Systems (Shanghai) Co., Ltd., headquartered in Shanghai, China. and peripheral components [20]. A 5 V–500 mA solar panel is integrated to support battery charging and ensure energy autonomy during prolonged outdoor deployment. This configuration enables continuous operation under variable sunlight conditions without an external power infrastructure.

#### 3.1.2. UV Measurement Transmitter Circuit

The LoRa Heltec ESP32 microcontroller plays a crucial role in data acquisition and processing and is distinguished by its ability to handle long-range communication via the LoRa protocol. In addition, the ESP32 offers multiple analog and digital ports for input and output, making it suitable for connecting various types of sensors. For a better understanding of its capabilities, the specifications of the LoRa Heltec ESP32 are shown in Table 2.

Regarding the microcontroller’s pin layout, the most notable pins are the VCC pin, GND pin, and OUT pin, which are essential for connecting and integrating the UV sensor. In Proteus, a package was developed that accurately replicates the connection between the LoRa Heltec ESP32 microcontroller and the UVM-30A ultraviolet radiation sensor, (generic module commonly distributed without a specified manufacturer, for this project, it was sourced from DFRobot, Shanghai, China) generating virtual models that faithfully reproduce the physical layout of the pins and the electrical characteristics of both devices. The ESP32 was modeled with all relevant pins, including those for LoRa communication and input/output ports, ensuring a precise and consistent simulation with its physical version. The flowchart in Figure 2 details each stage of the process, from data transmission to error management in ESP32 communication.

Before undertaking the integration of the circuit, an analysis was conducted on three primary components: the UV radiation sensor, the LEDs, and the power supply, as illustrated in Figure 3. The following is a detailed description of each component:The UVM-30A sensor, known for its accuracy, provides an analog output connected to the ESP32’s ADC, converting a 606 mV signal into a digital value of 752, corresponding to a level 6 solar radiation, using a 12-bit ADC (4096 levels).To enhance the visualization of UV radiation data, we integrated LEDs with differentiated colors to represent various levels of radiation, acting as a visual indicator similar to a traffic light. These LEDs, controlled by SONGLE SRD-05VDC-SL-C relays from the ESP32, provide a clear and visible representation, even in daylight conditions.To ensure a stable power supply, a 7805 voltage regulator was used, which converts a 12V input into a constant 5V output, complemented by capacitors and an indicator LED to guarantee the correct operation of the circuit.

#### 3.1.3. UV Measurement Receiver Circuit

The receiver, or gateway, uses the LoRa Heltec ESP32 V2 module in receiver mode, with the same power supply configuration as the transmitter. The data received are transmitted to an external platform through HTTP communication. For direct visualization, 8 × 8 MAX7219 LED matrices were integrated to display UV radiation levels on a numerical scale from 1 to 11, allowing users to access this information without the need to connect to the Ubidots platform, as shown in Figure 4.

#### 3.1.4. Calibration, Energy Management, and Node Deployment Strategy

Calibration of the UV sensors was performed by comparing the sensor readings with those obtained from certified professional radiometers used by the Quito Environmental Monitoring Network, Ecuador. The reference equipment is regularly calibrated to international standards. Discrepancies between the sensor data and the reference radiometers were recorded, and a correction function was implemented in the system’s microcontroller firmware to ensure accurate and consistent measurements in the field. The measurement node operates with an average energy consumption of approximately 75 mA during active measurement and LoRa transmission [22], which lasts around 2–3 s per cycle. In deep sleep mode, when the node is not performing measurements, the current consumption drops to less than 10 µA, optimizing energy usage for extended periods of operation without constant recharging. The node is powered by a 3.7 V lithium-polymer (LiPo) rechargeable battery with a capacity of 2500 mAh. This battery is supported by a 5 V–500 mA solar panel, ensuring energy autonomy and continuous operation even in remote off-grid environments. To further reduce power consumption, the ESP32 microcontroller is set to enter deep sleep mode between measurements. The system performs measurements every 10 min, adhering to the 1% duty cycle limitation imposed by the LoRaWAN protocol. Each measurement cycle, including data transmission, lasts only 100 ms before the node returns to deep sleep mode, optimizing the energy efficiency of the system. The node locations were chosen based on key criteria: direct exposure to sunlight to ensure accurate UV measurements and effective solar panel charging; verified LoRaWAN network coverage, confirmed through preliminary field tests with a strategically placed gateway; and accessibility for regular maintenance and to ensure the physical security of the equipment.

Table 3 summarizes the technical specifications outlined in the preceding section.

### 3.2. Implementation of the Solution

#### 3.2.1. Development of Transmitters on Protoboards

To initiate the transmitter prototype, protoboards were chosen to facilitate easy modifications and functional tests. Components such as the Heltec ESP32 LoRa microcontroller, the UVM-30A ultraviolet radiation sensor, and an ESP32 LoRa antenna with a nominal gain of 2 dBi were assembled, as specified in [23]. The prototyped transmitter circuit shown in Figure 5 was programmed using the Arduino environment due to its library support and ease of use, which allows efficient implementation of UV radiation data transmission functions.

The transmitter prototype includes an OLED display integrated with the LoRa module, which displays real-time UV radiation values before transmitting them to the gateway. This design ensures reliable data visualization on the OLED and validates data reception by the gateway, as shown in Figure 6.

#### 3.2.2. Development of the Receiver (Gateway)

Subsequently, a receiver or gateway was developed on a protoboard for initial testing and adjustments. Using components such as the LoRa Heltec ESP32, an ESP32 LoRa antenna, and a 12 V power supply, the prototype was assembled, as shown in Figure 6.

The gateway program enables the reception of UV radiation data from the transmitters, displays it on its OLED screen, and sends it to the Ubidots platform via HTTP communication. Within the module’s programming, libraries such as ‘SPI.h’, ‘LoRa.h’, ‘SSD1306.h’, and ‘UbidotsEsp32Mqtt.h’ were used to enable LoRa protocol communication, SPI functionality, OLED management, and connectivity to Ubidots through MQTT.

#### 3.2.3. Migration to PCBs

To enhance durability and performance stability, the circuits were migrated from the protoboards to printed circuit boards (PCBs) to provide efficient integration, a compact size, and improved reliability for data collection and transmission, as verified during testing and shown in Figure 7.

#### 3.2.4. Additional Elements

In addition to transmitters and receivers, additional elements, such as protective structures and visual aids, were incorporated to enhance the reliability and durability of the system. The QPLITE EDMP-12 LED model, operating at 50/60 Hz with a controlled voltage input range of 85–265 V, was selected for its robustness and visual feedback. These LEDs, placed on the transmitters, use red, yellow, and green indicators to represent various radiation levels, while the 8 × 8 LED matrices on the gateway allow for quick indoor visualization of radiation levels, as shown in Figure 8, Figure 9 and Figure 10.

To protect electrical devices from adverse weather conditions, physical structures were implemented to house the transmitters, LEDs, and receivers with their LED matrices. Materials such as steel were used for the structures and vinyl for the visual elements, while a simpler plastic structure was used for the receiver or gateway, as shown in Figure 11 and Figure 12.

As mentioned above, to display real-time UV radiation data, alert users for high levels, and store information, the system is connected to the Ubidots platform. To start, an account must be registered on their website, a new project created, and the ESP32 microcontroller selected, thus generating a unique ID and API token to facilitate communication.

In receiver programming, the connection to Ubidots is configured using its library, first entering the WiFi network credentials, followed by the ID and API token. This enables the ESP32 to connect to the network and send data, including two variables of UV radiation from the two solarimeters, as shown in Figure 13.

Finally, on the main Ubidots dashboard, widgets are selected to visualize UV radiation levels, receive alerts if they exceed safe thresholds, and determine the type of data to be stored, as shown in Figure 14 and Figure 15.

### 3.3. Testing and Evaluation of the Prototype

At this stage, tests for measuring UV radiation were conducted using LoRa communication at a frequency of 433 MHz to ensure continuous range and transmission. The data were stored in the cloud and validated using the official environmental monitoring system of the Secretary of Environment of the Metropolitan District of Quito, which measures UV radiation as one of its parameters.

#### 3.3.1. UV Radiation Tests

To validate the UV radiation levels of the UVM-30A sensor, tests were conducted under various climatic conditions, as shown in Figure 16. The initial tests, carried out on the protoboards, indicated that at 2:30 p.m. on 7 June 2024, a radiation level of 5 was recorded near the University of the Americas, confirming the UV data collection.

A UV radiometer (Biospherical Instruments GUV2511) from the Ministry of Environment was used as a reference, located in Jipijapa, Quito, less than 3 km from the testing sites, as shown in Figure 17. At the same date and time, this radiometer also recorded levels close to 5, which corresponded to the data from the prototype.

Subsequently, the test was replicated with the circuit on a PCB, yielding similar results but showing a notable improvement in stability and accuracy compared to the protoboard tests, as shown in Figure 18 and Figure 19. These results validated the effectiveness of the UV30 sensor compared to the official radiometer.

#### 3.3.2. Distance Tests with Heltec LoRa Modules

The next phase of the test focused on LoRa communication, assessing the range and continuous transmission. The trials were carried out in a controlled environment at the Universidad de las Américas, where effective communication between the transmitters and receivers on the protoboards was achieved, reaching up to 220 m, as shown in Figure 20. At this distance, the signal exhibited an attenuation of 88.07 dB.

With the circuit implemented on the PCBs, communication improved significantly, reaching 335 m with an attenuation of 91.59 dB, as shown in Figure 21. This increase is consistent with the expected behavior of electromagnetic waves, where signal loss increases with distance.

#### 3.3.3. Radio Propagation Modeling and Simulation

To complement the empirical measurements conducted in the field, a series of simulations were performed using the Radio Mobile tool, which is based on the Longley–Rice propagation model. This tool enables the prediction of terrain-aware radio coverage by incorporating elevation data (SRTM) and environmental conditions.

The simulations were conducted to replicate and analyze signal behavior under the real topographical and environmental conditions of the ecological trail. Each scenario was designed using georeferenced profiles and actual device parameters, such as frequency (433 MHz), antenna heights, and transmission power, to model the expected path loss and received power over various terrain configurations. The goal was to identify potential coverage limitations, assess line-of-sight (LoS) availability, and compare the simulated results with in situ measurements to determine the impact of vegetation, terrain irregularities, and elevation profiles on LoRaWAN signal performance.

This simulation approach provides a predictive baseline to contrast with real-world behavior, especially in conditions where dynamic losses (e.g., multipath or vegetation absorption) are difficult to fully capture using theoretical models alone. Scenarios were selected to represent both optimal and suboptimal propagation conditions, including clear terrain, forested sections, and elevation transitions near a ravine. Table 4 describes the simulated scenarios, and Table 5 provides the technical details of the simulation using Radio Mobile. For all simulation scenarios, the transmitter coordinates and parameters were the latitude (Tx): −0.162929; longitude (Tx): −78.45917; transmit power (dBm): 15; and receiver sensitivity (dBm): −124.

**Scenario 1:** A section of the ecological trail with a slope of 4° was simulated using SRTM elevation data in Radio Mobile. The transmitter and receiver antennas were set at 2 m and 1 m, respectively, over a distance of 336 m. The goal was to assess the LoS conditions and estimate the path loss. The results showed a path loss of 102.6 dB and a received power of −84.6 dBm, which is well above the sensitivity threshold (−124 dBm), indicating a functional link. However, significant signal degradation was evident due to the inclined terrain. This effect can be observed in Figure 22.

**Scenario 2:** A simulation was conducted in Radio Mobile to replicate the real conditions that caused a 92 dB signal at 336 m. The setup used a higher transmitter height (10 m) to analyze the impact of the terrain, antenna height, and 433 MHz frequency on link performance. Despite a theoretical received power of −73.8 dBm with a simulated path loss of 91.8 dB, the link failed in practice. This suggests the presence of additional unmodeled losses, such as vegetation absorption, destructive multipath, or environmental interference. Although the receiver sensitivity was −124 dBm, the actual sensitivity depends on the spread factor (SF) and bandwidth (BW); default settings prioritizing low latency or energy savings may reduce effective sensitivity, increasing demodulation errors, and contributing to communication loss. This effect can be observed in Figure 23.

**Scenario 3:** LoRaWAN communication was evaluated over a distance of 370 m in an environment characterized by natural vertical obstructions, where the receiver was located in a low-lying area near a ravine, at a negative relative elevation compared to the transmitter, and surrounded by dense vegetation. This configuration resulted in significant signal degradation, with a path loss of 106.2 dB and a received signal strength of −88.2 dBm. Although the signal remained above the sensitivity threshold, it approached a low link margin. Sloped terrain, ground humidity, and dense vegetation contributed significantly to signal attenuation, demonstrating how topographic and environmental factors can severely impair the propagation of 433 MHz signals, even over relatively short distances. This scenario can be observed in Figure 24.

**Scenarios 4 and 5:** These scenarios enabled a comparative analysis between an open area and a forested area, maintaining a constant transmission distance (490 m) and antenna heights (10 m at the transmitter and 2 m at the receiver). Two profiles were selected: one located on a clear terrain segment and another within a densely vegetated ecological trail. The objective was to demonstrate the direct impact of the surrounding environment on the propagation loss, highlighting how the presence of vegetation can increase the path loss and degrade link performance, despite identical geometric conditions. Both cases yielded identical simulation results, with a path loss of 88.7 dB and received power of −70.7 dBm, suggesting that vegetation had no significant effect under the simulated model. However, real-world deployments may differ. In particular, the elevation of the receiver, which was comparable to that of the transmitter, considerably reduced the propagation loss, indicating that favorable topography mitigates the impact of distance. These scenarios can be observed in Figure 25 and Figure 26.

**Scenario 6:** This scenario focused on an open area adjacent to a forested area, using a link distance of 560 m and identical antenna heights enabled by the topographic profile. This setup aimed to evaluate how environmental characteristics, particularly dense vegetation, as found along the ecological trail, influence the propagation loss. By keeping the technical parameters constant, the effect of natural clutter was isolated, showing that even with equivalent geometric conditions, vegetated environments can increase the path loss and compromise link quality. The simulation yielded a path loss of 93.3 dB and a received power of −75.3 dBm. Although the signal remained within the link margin, the conditions were more demanding. This scenario confirms that, even at longer distances, favorable alignment and relative visibility can sustain the link despite environmental obstacles. These effects can be observed in Figure 27.

#### 3.3.4. Tests with the Ubidots Platform

The proper functioning of the Ubidots platform is crucial for visualization, data storage, and notifications about elevated levels of UV radiation. To ensure its effectiveness, various tests were conducted.

First, it was verified that the radiation levels from each UV meter were displayed in real time in their respective dashboard widgets. As shown in Figure 28, the UV radiation level reached a value of 5 at 2:30 p.m. on 7 July 2024, which coincides with the readings taken on the protoboards and confirms the precision of the data reflected on the platform (see Figure 18 and Figure 19). A minimum data update frequency was configured, and the bar chart widget was scheduled to update every 10 min, allowing the monitoring of radiation trends.

Additionally, the capability of the platform to store historical data was evaluated. By activating the “timestamp” feature, the data were securely saved in the “Events” section, enabling comparisons with other UV radiation data sources. It was decided to record data twice per hour, in accordance with the limitations of the free version of Ubidots.

Finally, the system alerts, which notify users of dangerous levels of UV radiation, were tested. Maximum thresholds were established, and when these thresholds were exceeded, automatic alerts were generated, facilitating prompt decision-making.

#### 3.3.5. Estimated Duty Cycle Calculation

To estimate the duty cycle, the following assumptions were made:Transmission frequency: **6 times per hour.**Approximate duration of each packet: **100 ms** (conservative estimate based on the initial LoRa configuration).
$$\begin{array}{cc} \mathrm{Total}\;\mathrm{transmission}\;\mathrm{time}\;\mathrm{per}\;\mathrm{hour} & =6\times 100\;\mathrm{ms}=600\;\mathrm{ms}\\ \mathrm{Total}\;\mathrm{time}\;\mathrm{in}\;\mathrm{one}\;\mathrm{hour} & =3600\;s=3,600,000\;\mathrm{ms}\\ \mathrm{Duty}\;\mathrm{cycle} & =\frac{600\;\mathrm{ms}}{3,600,000\;\mathrm{ms}}\approx 0.0167\% \end{array}$$

The results showed that the estimated duty cycle was approximately **0.0167%**, which is significantly below the maximum allowed limit of **1%** for ISM band operations, as specified by the ETSI EN 300 220-2 V3.2.1 (2018-06) standard [25].

### 3.4. Results and Discussion

A data collection period was established for testing purposes due to the limited storage available in the free version of Ubidots for meteorological data. In addition, the system was configured to operate from 8 a.m. to 6 p.m. and then enter ‘deep sleep’ mode to conserve energy and comply with ITU recommendations. This setup also aimed to replicate the monitoring system used by the Environmental Secretariat in Quito and ensure data accuracy. The results are presented below.

#### 3.4.1. Comparison of UV Radiation Recorded by Each Individual Solarimeter

To evaluate the individual performance of each solarimeter, the data collected on Tuesday, 2 July, and Wednesday, 3 July, were analyzed. The results showed consistent radiation readings between both devices, except at 3:00 p.m. on Tuesday, when the first solarimeter recorded a level of 4 and the second a level of 1, reflecting a difference of 3 units, as shown in Figure 29. After monitoring the second solarimeter the following day, it was found that the shadow of a nearby column in the northeast courtyard of the Universidad de las Américas affected its readings. To correct this, the solarimeter was relocated out of the shadow’s range, thus ensuring the reliability of the UV radiation data in the area.

#### 3.4.2. Data Collection and Analysis

The first step was to validate these data by comparing them with those recorded by the Environmental Secretariat during the same period. The comparison revealed consistent results, as seen in Figure 30, taking into account operating hours and the equipment used, which supports the reliability of the measurements.

The analysis indicated that UV radiation levels increased exponentially from 10:00 a.m., reaching their peak between 11:00 a.m. and 2:00 p.m., as illustrated in Figure 2, which presents the average UV radiation by hour. Although UV radiation followed a general pattern, significant variations were observed on specific days and hours, reaching health-threatening levels. Six alerts were issued, primarily between 11:00 a.m. and 2:00 p.m., as shown in Figure 31 and Figure 32.

#### 3.4.3. Prototype Accuracy Compared to Other Professional UV Radiation Meters

The data obtained from the solarimeters were compared with those collected by the receiver or gateway and the radiometers recorded by the Environmental Secretariat, aiming to assess the accuracy of the implemented prototype. This was achieved by computing the percentage error, as demonstrated in the calculation below. The percentage error is quantified using the following expression:(1)$${\%}_{error}=\frac{Xprofessional-Xrecorded}{Xprofessional*100},$$
where *X*professional corresponds to the specific level measured by the radiometer from the Environmental Secretariat and *X*recorded is the level registered by the prototype. For the calculation, the following values recorded on 4 July 2024 were used: the value recorded by the Environmental Secretariat at 10:40 a.m., which was 7.5, as shown in Figure 33.

Figure 34 shows that the value recorded by the prototype at 10:40 a.m. was 7.3.

Applying these values within the context of Equation (Equation 1), the resulting expression is derived as follows:$${\%}_{error}=\frac{7.5-7.3}{7.5*100},$$(2)$${\%}_{error}=2.67$$

The correlation between the data collected by the solarimeters and the radiometers showed a discordant value of 3%. Based on these data, it can be concluded that there is a strong relationship between the two UV radiation records, with a correlation coefficient of 0.97. This result demonstrates the precision of the prototype in measuring UV radiation, meeting the required levels of precision necessary for both health-related applications and environmental monitoring.

## 4. Conclusions

An analysis of the architecture and components of LoRaWAN technology enabled the design of an effective solution for monitoring UV radiation, considering the operating frequencies recommended by the ITU and compatible devices. The prototype demonstrated its efficiency during validation on the protoboards and PCBs, as well as in its implementation at Universidad de las Américas, where the testing revealed only a 3% discrepancy compared to professional systems, demonstrating its reliability.

The collected data revealed significant patterns in UV radiation levels, showing an exponential increase between 11:00 a.m. and 2:00 p.m., a critical period for protecting the health of students and staff. Effective integration with the Ubidots platform facilitated data visualization and storage, allowing students to monitor elevated UV radiation levels. These results highlight the positive impact of the implemented solution at the Universidad de las Américas.

The precision of the prototype was validated against the records of the Environmental Secretariat, showing a correlation of 97% and a percentage error of only 2.67%. This shows that the prototype meets the required precision levels for health and environmental monitoring applications, offering a reliable and accessible alternative to professional systems.

During implementation, technical challenges were encountered in LoRa communication, particularly in terms of continuous transmission and signal range. In controlled tests within the Universidad de las Américas, effective communication between the transmitters and receivers on the protoboards was achieved up to 220 m, with an attenuation of 88.07 dB. However, with their implementation on the PCBs, the range improved significantly to 335 m, with an attenuation of 91.59 dB, confirming the expected relationship between distance and signal loss.

Although simulations in Radio Mobile provided a solid technical foundation, real-world environmental conditions, such as vegetation, humidity, multipath propagation, and irregular topography, could introduce significant additional losses that may not always be captured by the model, potentially leading to link failure even before reaching the theoretical sensitivity limit. It was confirmed that transmitter elevation significantly enhanced link performance, particularly in sloped terrain and vegetated environments. Furthermore, the area adjacent to the ravine represents a critical environment that requires experimental validation, while scenarios with favorable transmitter positioning demonstrated the potential to achieve greater distances, even in the presence of light vegetation.

With regard to scalability and sustainability, the findings suggest that the solution can be expanded to other areas or institutions. Future work should consider installing at least five additional devices in the Quito Metropolitan District, allowing data concentration on the Ubidots platform. This would facilitate public access to real-time UV radiation data via mobile devices or web-based applications.

Finally, there are significant opportunities for improvement and innovation based on the results obtained. A key next step would be to optimize the remote connectivity of modules in different locations within Quito, as well as develop a cloud-based platform that centralizes the data and presents them in a user-friendly format for web and mobile applications. These enhancements would not only expand the reach of the system but also strengthen its impact on education, research, and public health decision-making.

## Figures and Tables

**Figure 1 sensors-25-03110-f001:**
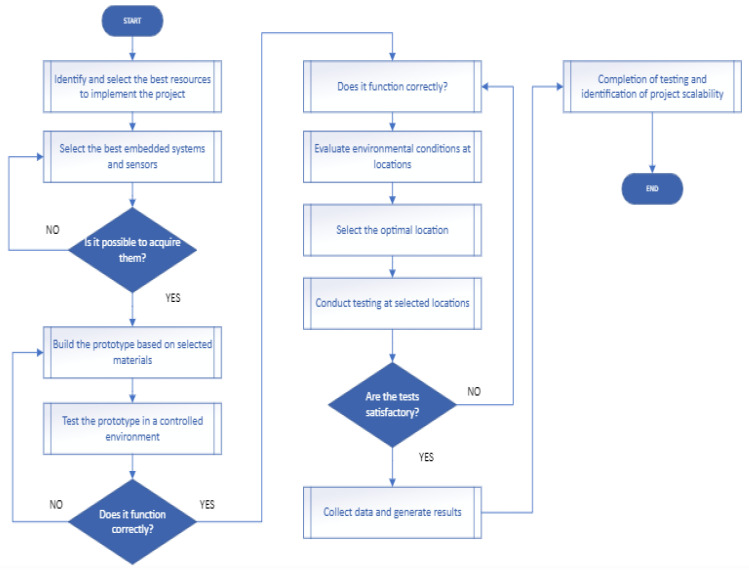
Flowchart of the solution development process.

**Figure 2 sensors-25-03110-f002:**
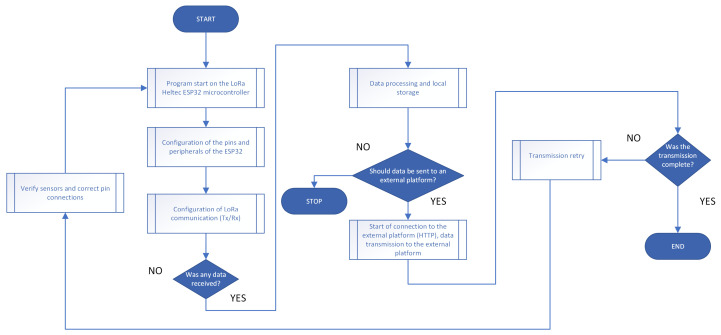
Flowchart of the LoRa Heltec ESP32 microcontroller operation.

**Figure 3 sensors-25-03110-f003:**
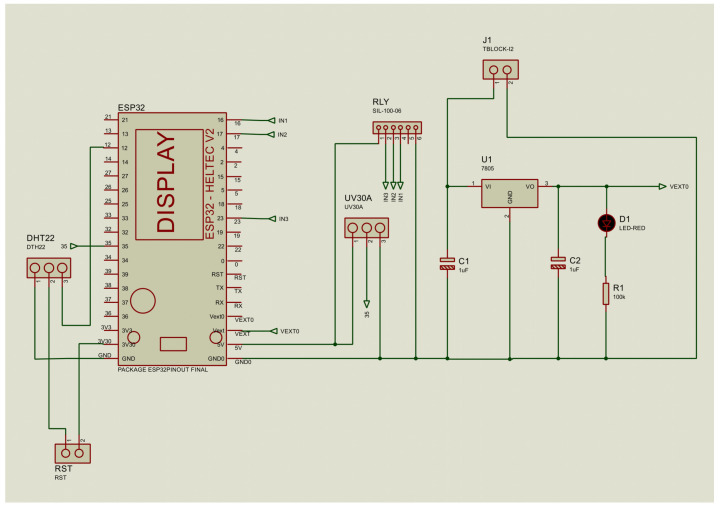
Structure of the voltage regulator alongside the LoRa Heltec ESP32 microcontroller and UVM-30A sensor on the Proteus platform.

**Figure 4 sensors-25-03110-f004:**
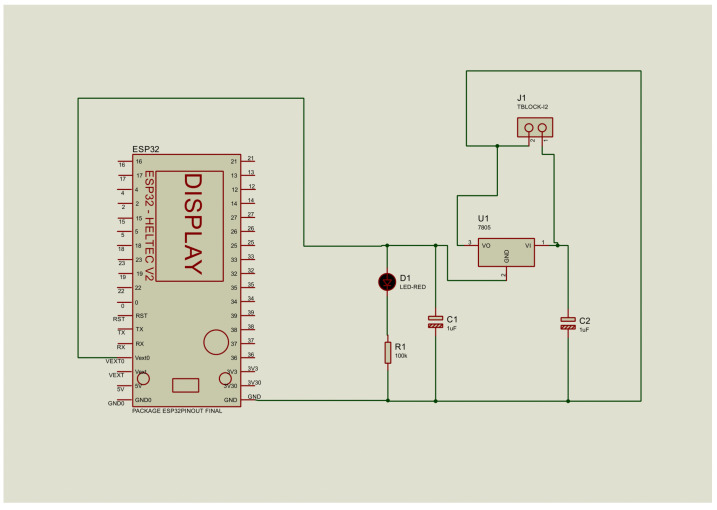
Design of the receiver circuit on the Proteus platform.

**Figure 5 sensors-25-03110-f005:**
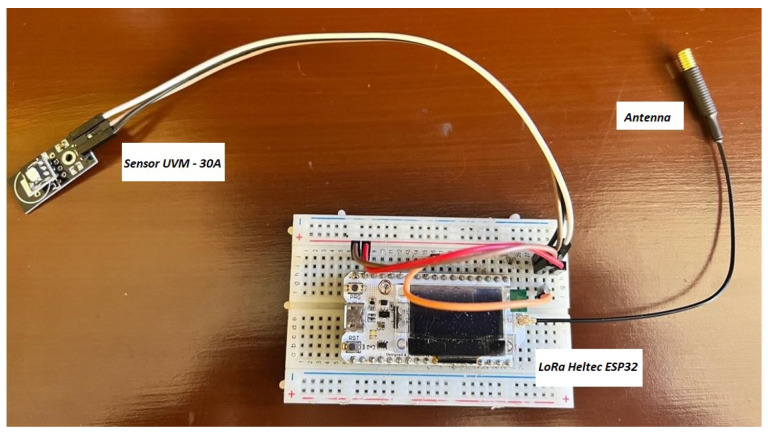
Transmitter circuit on a protoboard.

**Figure 6 sensors-25-03110-f006:**
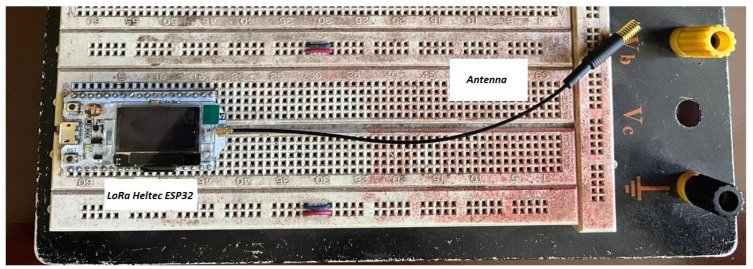
Receiver or gateway circuit in operation on a protoboard.

**Figure 7 sensors-25-03110-f007:**
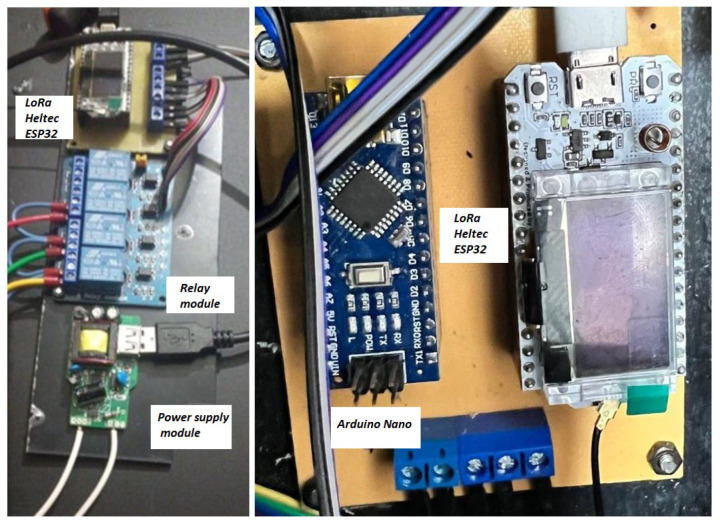
Prototypes on PCBs.

**Figure 8 sensors-25-03110-f008:**
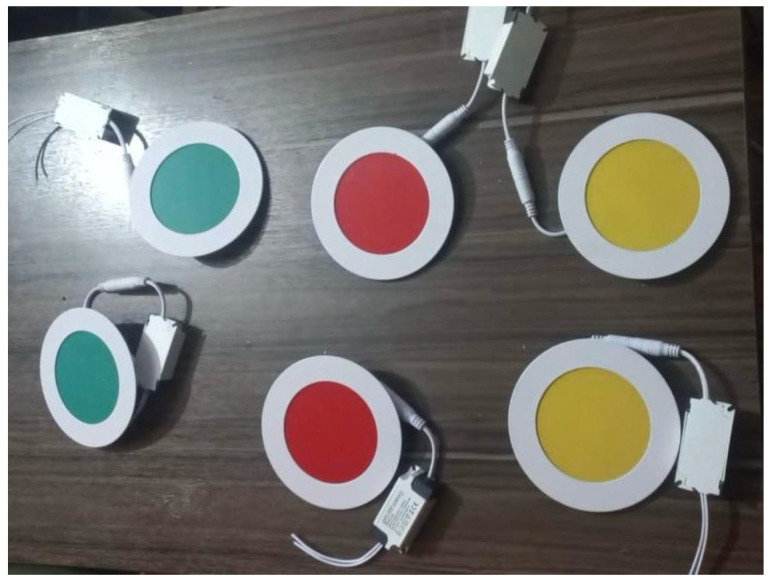
LEDs to indicate radiation levels.

**Figure 9 sensors-25-03110-f009:**
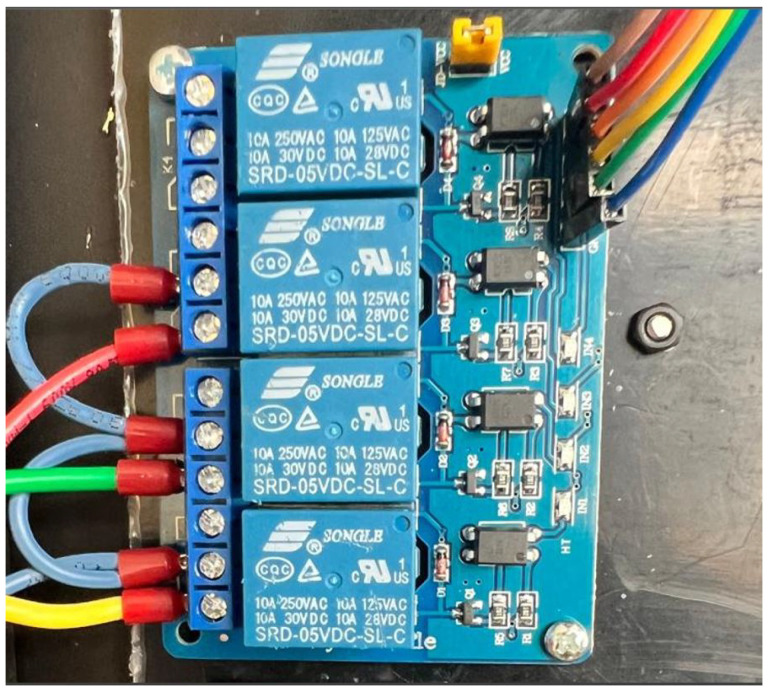
Connection of LED wires to the relays.

**Figure 10 sensors-25-03110-f010:**
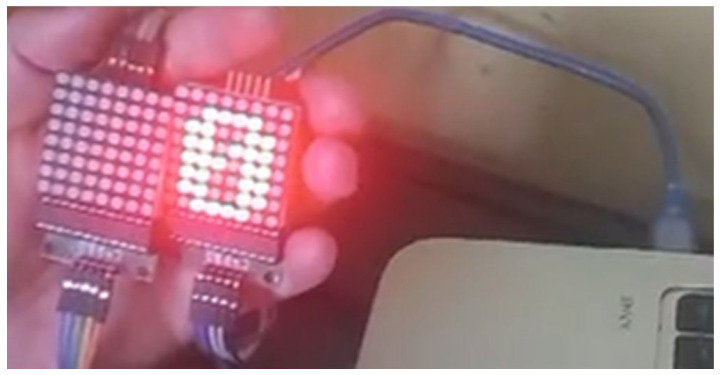
LED matrix on the receiver or gateway.

**Figure 11 sensors-25-03110-f011:**
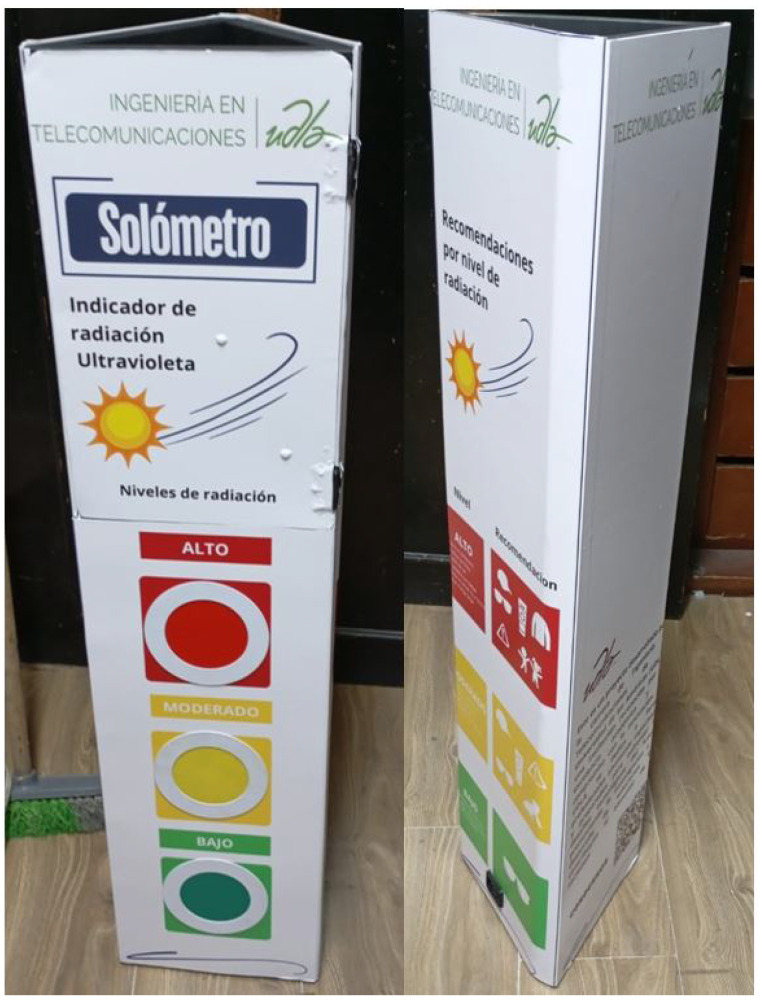
Qualitative LED matrix on the receiver or gateway.

**Figure 12 sensors-25-03110-f012:**
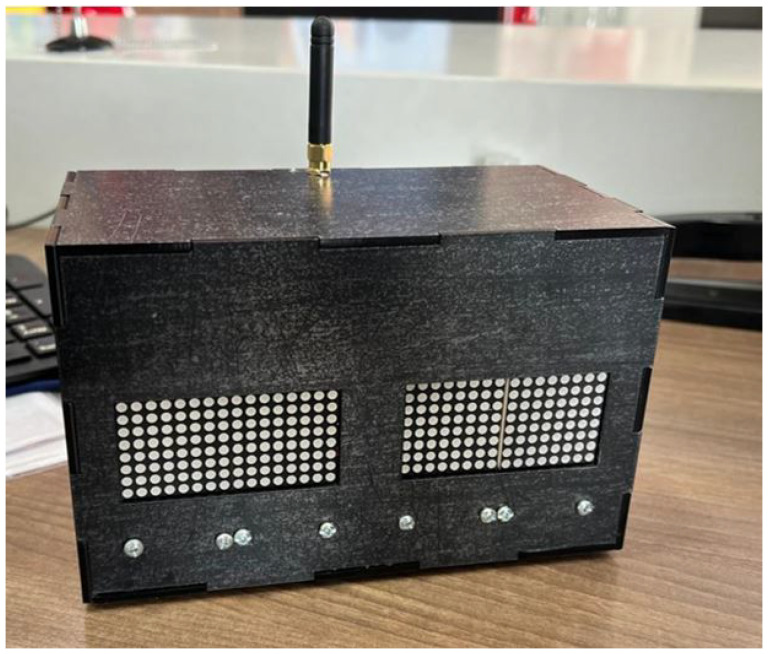
Quantitative matrix of the LEDs on the receiver or gateway.

**Figure 13 sensors-25-03110-f013:**
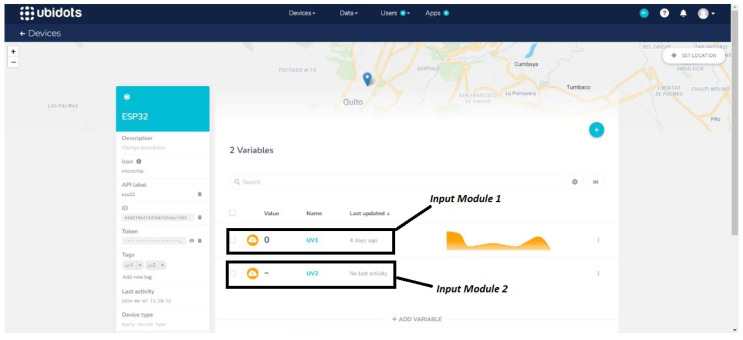
Selection of the number and type of variables in Ubidots.

**Figure 14 sensors-25-03110-f014:**
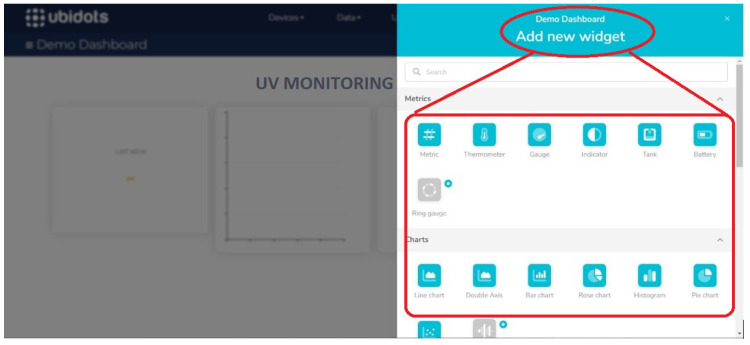
Selection of widgets.

**Figure 15 sensors-25-03110-f015:**
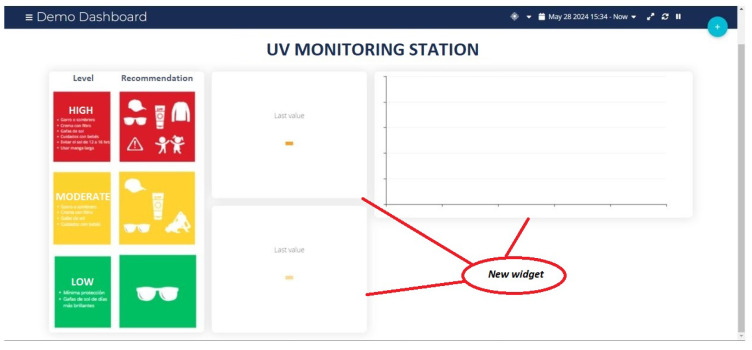
Selected widgets on the main dashboard of Ubidots.

**Figure 16 sensors-25-03110-f016:**
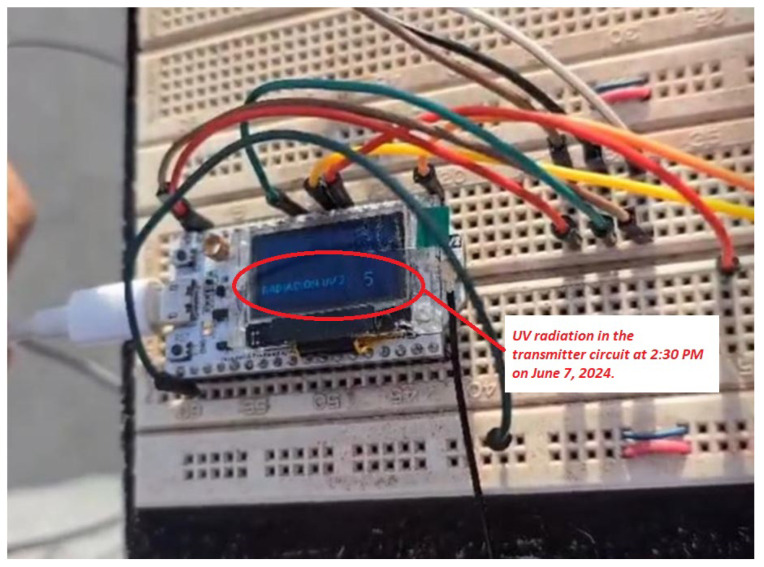
UV radiation in the transmitter circuit on a breadboard at 2:30 p.m. on 7 June 2024.

**Figure 17 sensors-25-03110-f017:**
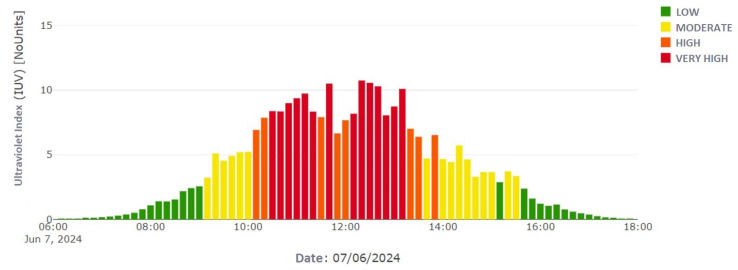
Bar graph showing UV radiation records provided by the Ministry of Environment’s radiometer for 7 June 2024 [24].

**Figure 18 sensors-25-03110-f018:**
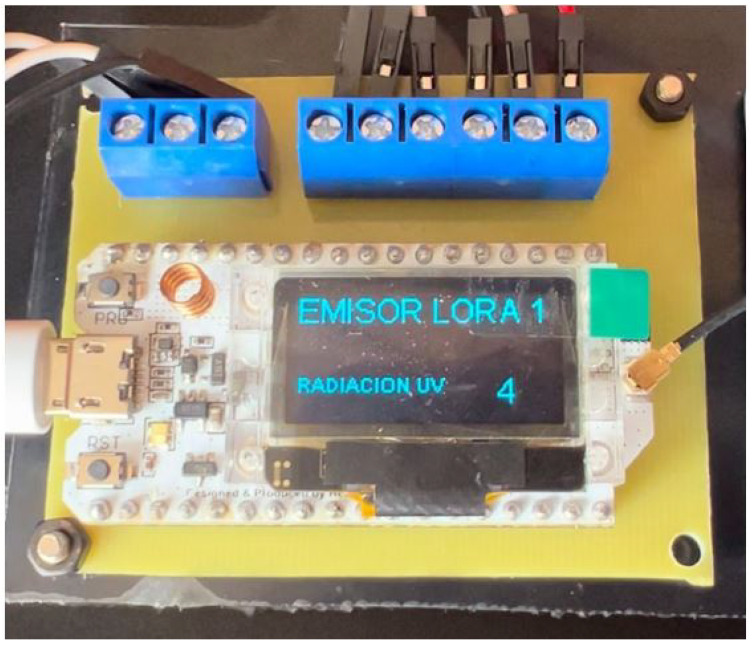
UV radiation in the transmitter circuit on a PCB at 3 p.m. on 12 June 2024.

**Figure 19 sensors-25-03110-f019:**
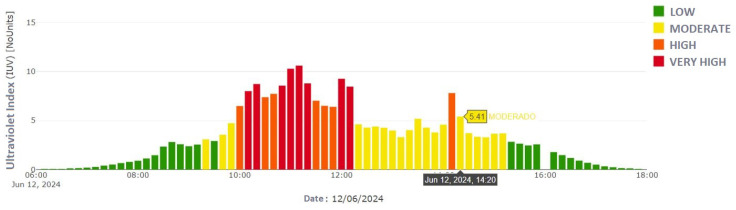
Bar graph of UV radiation records provided by the Ministry of Environment’s radiometer for 12 June 2024 [24].

**Figure 20 sensors-25-03110-f020:**
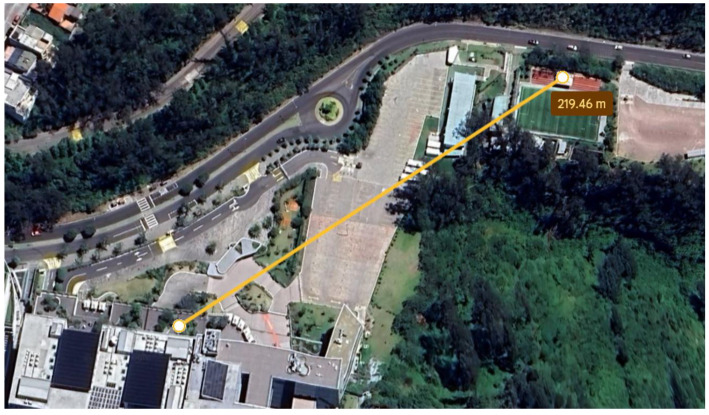
Maximum distance achieved with the prototypes on the protoboards.

**Figure 21 sensors-25-03110-f021:**
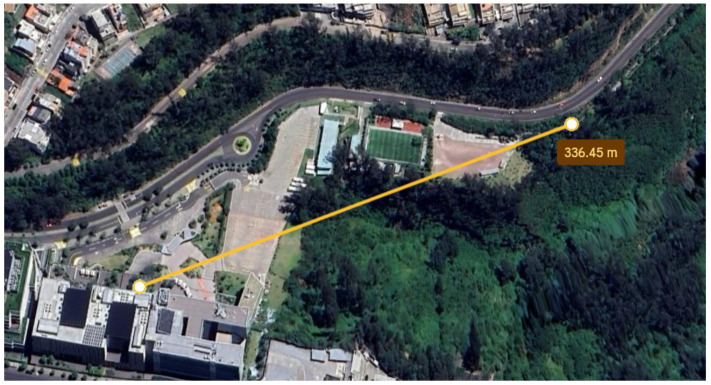
Maximum distance achieved with the prototypes on the PCBs.

**Figure 22 sensors-25-03110-f022:**
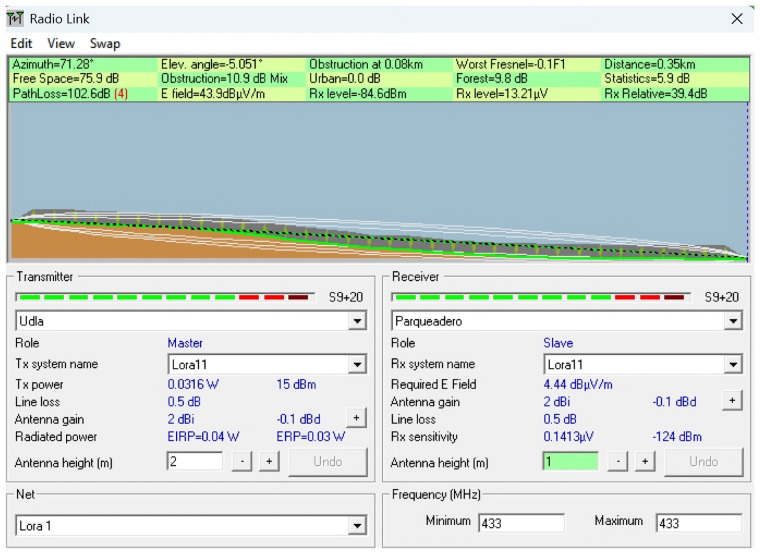
Simulation of Scenario 1: Line-of-sight conditions and path loss on a 4-degree slope section of the ecological trail using elevation data in Radio Mobile.

**Figure 23 sensors-25-03110-f023:**
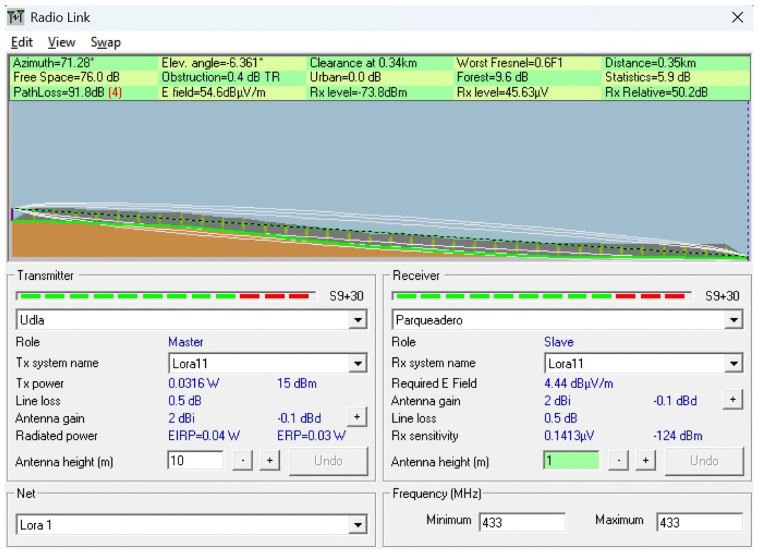
Simulation of Scenario 2: Terrain and antenna height effects on link performance at 433 MHz over 336 meters with observed signal loss.

**Figure 24 sensors-25-03110-f024:**
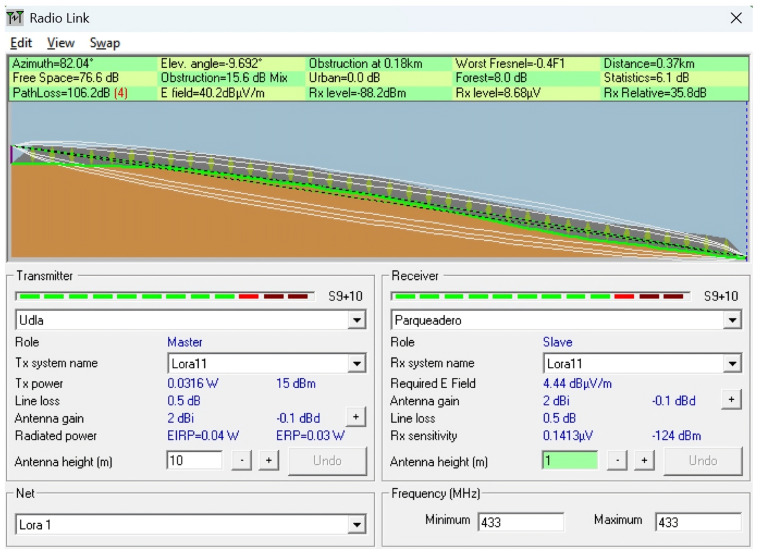
Simulation of Scenario 3: Evaluation of 433 MHz signal degradation over 370 meters due to terrain slope, dense vegetation, and negative receiver elevation.

**Figure 25 sensors-25-03110-f025:**
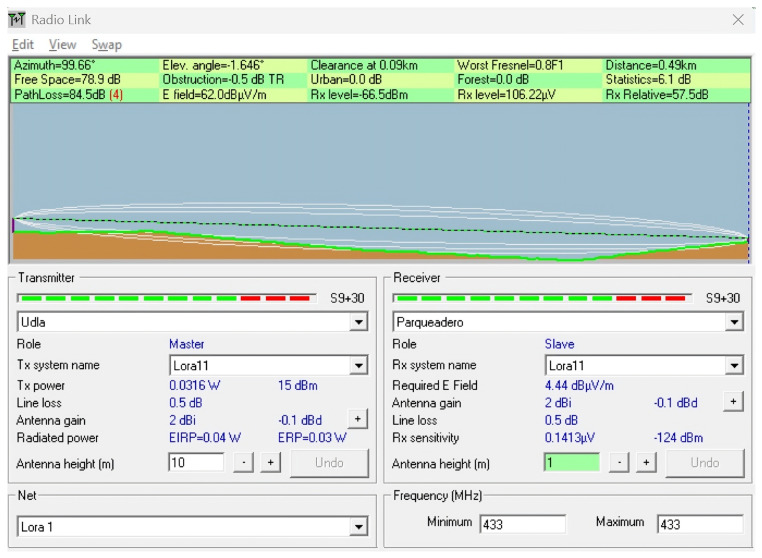
Simulation of Scenario 4: Simulated 433 MHz signal propagation over 490 meters in an open area with favorable topography.

**Figure 26 sensors-25-03110-f026:**
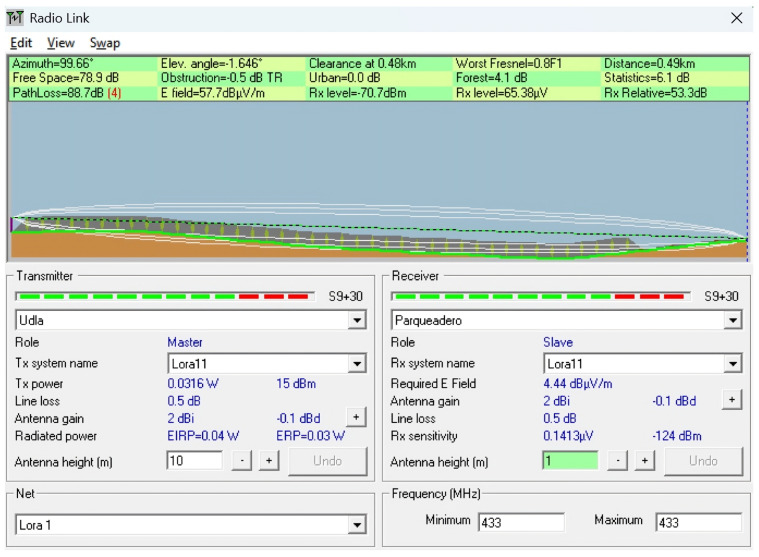
Simulation of Scenario 5: Simulated 433 MHz signal propagation over 490 meters in a densely vegetated area with comparable elevation.

**Figure 27 sensors-25-03110-f027:**
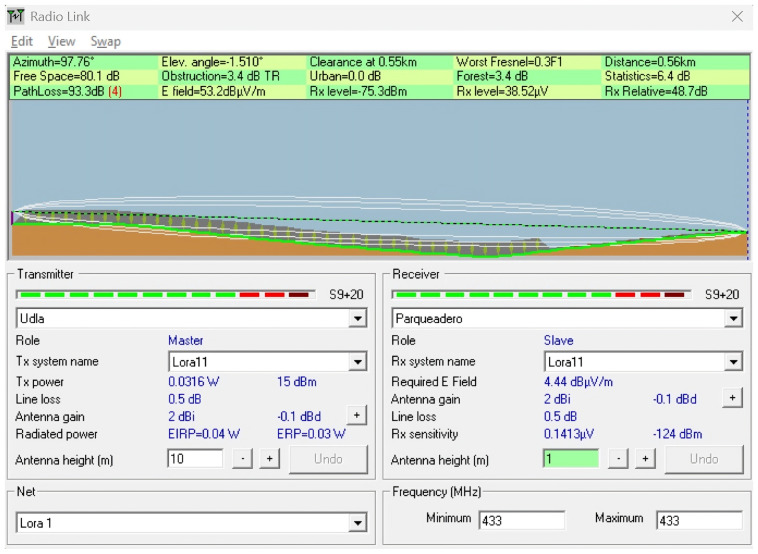
Simulation of Scenario 6: Simulation of 433 MHz signal propagation over 560 meters near a forested area to assess vegetation impact under aligned topographic conditions.

**Figure 28 sensors-25-03110-f028:**
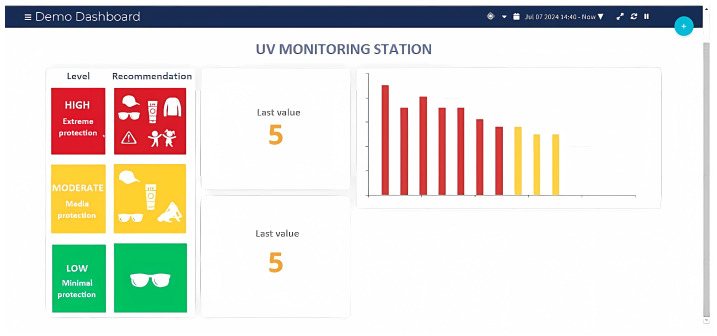
UV radiation at 2:30 p.m. on 7 June 2024, as shown on the Ubidots platform. UV radiation levels: green (low risk), orange (high risk), and red (very high risk).

**Figure 29 sensors-25-03110-f029:**
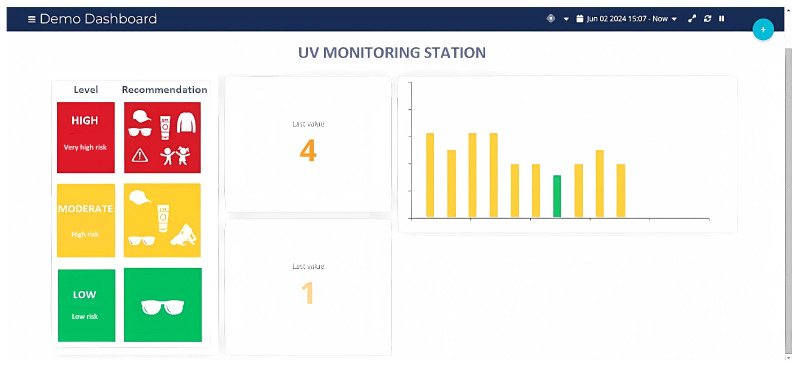
UV radiation at 3:00 p.m. on 2 July 2024, as shown on the Ubidots platform. UV radiation levels: green (low risk), orange (high risk), and red (very high risk).

**Figure 30 sensors-25-03110-f030:**
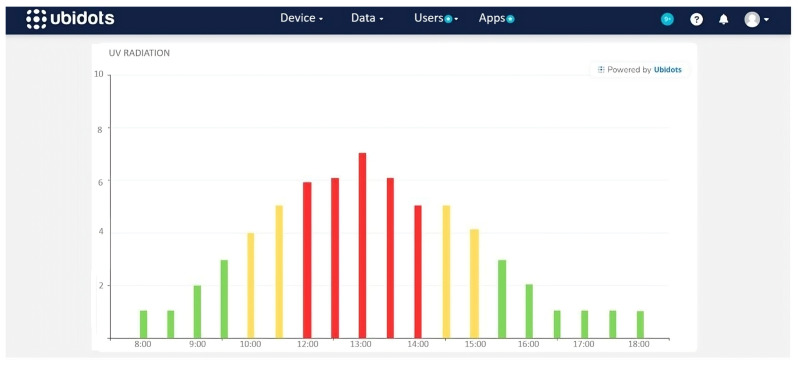
Bar chart showing the average UV radiation during the testing period.

**Figure 31 sensors-25-03110-f031:**
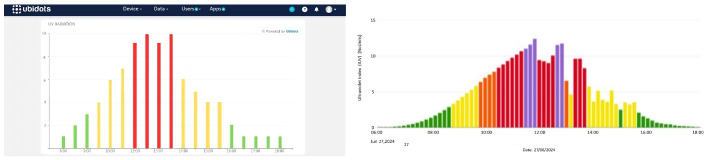
Variationsrecorded on 27 June 2024. Note: Data compared with the [24]. UV radiation levels (Left graph): green (low risk), orange (high risk), and red (very high risk). UV radiation levels (Right graph): green (low risk), yellow (moderate risk), orange (high risk), red (very high risk), and violet (extreme risk).

**Figure 32 sensors-25-03110-f032:**
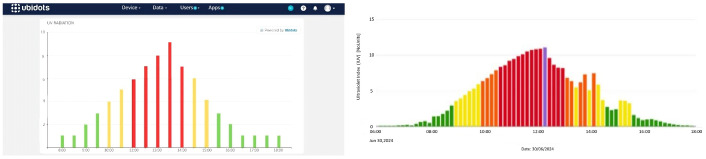
Variations recorded on 30 June 2024. Note: Data compared with the [24]. UV radiation levels (Left graph): green (low risk), orange (high risk), and red (very high risk). UV radiation levels (Right graph): green (low risk), yellow (moderate risk), orange (high risk), red (very high risk), and violet (extreme risk).

**Figure 33 sensors-25-03110-f033:**
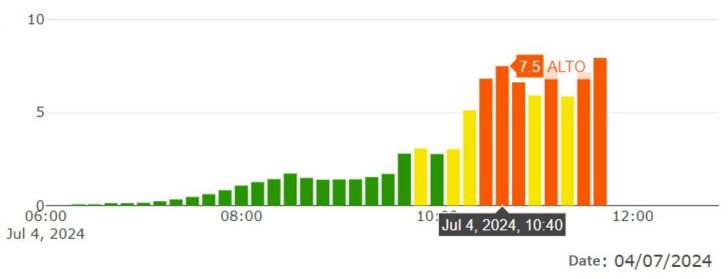
Value of radiation recorded by the Environmental Secretariat at 10:40 a.m. on 4 July 2024. Taken from the [24]. UV radiation levels: green (low risk), orange (high risk), and red (very high risk).

**Figure 34 sensors-25-03110-f034:**
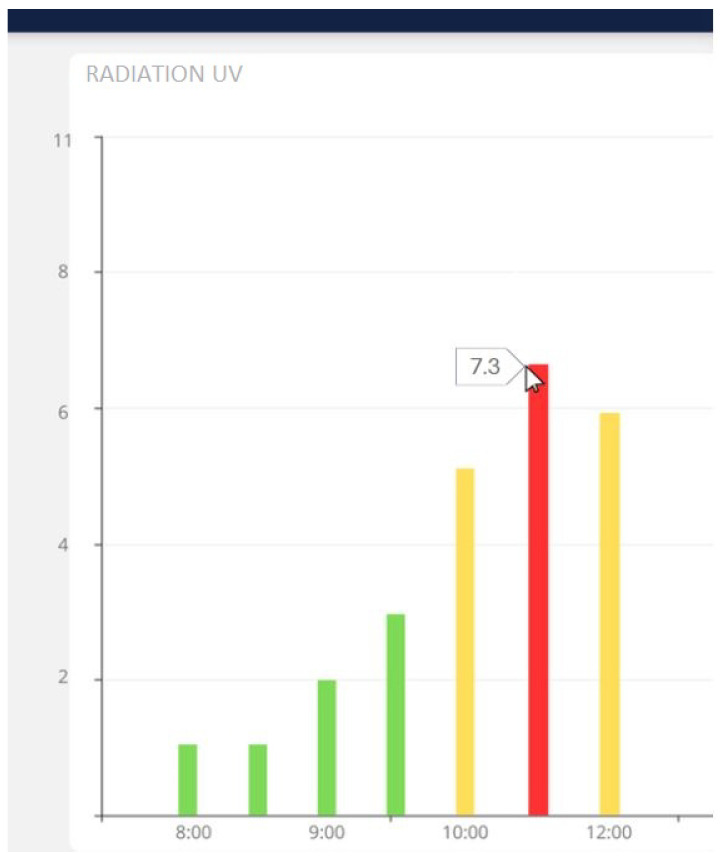
Value of radiation recorded by the prototype at 10:40 a.m. on 4 July 2024. UV radiation levels: green (low risk), orange (high risk), and red (very high risk).

**Table 1 sensors-25-03110-t001:** Qualitative SWOT analysis of the proposed monitoring solutions.

Proposal	Strengths	Weaknesses	Opportunities and Threats
**Nanosatellite-based Monitoring**	- High-precision data - Extensive coverage - Scientific value	- High cost and complexity - Launch and control limitations	*Opportunities:* International collaboration, high-impact research *Threats:* Launch failure, obsolescence, funding limitations
**UV Radiometers with Traditional Telemetry**	- Reliable and accurate data - Suitable for benchmark use	- Expensive equipment - Requires skilled maintenance	*Opportunities:* Use in industry or research labs *Threats:* Poor scalability, limited integration with LPWAN
**Solarimeters with LoRaWAN (Selected)**	- Cost-effective - Easy to deploy - Wireless long-range communication	- Lower precision than radiometers - Dependent on local gateways	*Opportunities:* Scalable for urban and rural networks, educational use *Threats:* Environmental interference, requires calibration

**Table 2 sensors-25-03110-t002:** Specifications of the LoRa Heltec ESP32 microcontroller, taken from [21].

Parameter	Description
**Master Chip**	ESP32-S3FN8 (Xtensa®32-bit lx7 dual-core processor)
**LoRa Chipset**	SX1262
**USB-to-Serial Chip**	CP2102
**Frequency**	470~510 MHz, 863~928 MHz
**Max. TX Power**	21 ± 1 dBm
**Max. Receiving sensitivity**	−134 dBm
**Wi-Fi**	802.11 b/g/n, up to 150 Mbps
**Bluetooth**	Bluetooth LE: Bluetooth 5, Bluetooth mesh

ESP32 microcontroller manufactured by Espressif Systems (Shanghai) Co., Ltd., headquartered in Shanghai, China.

**Table 3 sensors-25-03110-t003:** Technical specifications and implementation details of the UV monitoring system.

Aspect	Description
Sensor sensitivity	Calibrated UV sensors with a typical sensitivity of ±1 µW/cm^2^
Measurement range	0 to 15 mW/cm^2^
Accuracy	Validated in the laboratory by comparing against a certified professional radiometer from the Quito Environmental Monitoring Network
Power supply	A 3.7 V LiPo battery with a 2500 mAh capacity, combined with a 5 V–500 mA solar panel for energy autonomy
Average energy consumption	75 mA during active mode (lasting 2–3 s per cycle); below 10 µA in deep sleep mode
Energy-saving strategies	Deep sleep mode of the ESP32 microcontroller, cyclic measurement every 10 min, short LoRa transmission duration (100 ms)
Calibration process	Calibration was conducted by comparing the sensor’s readings with certified equipment from the municipal UV monitoring network; a correction function was applied in the firmware
Node placement criteria	Nodes were placed along an ecological trail on the university campus, considering (a) direct sunlight exposure; (b) verified LoRaWAN coverage from a strategically located gateway; and (c) physical safety and accessibility for periodic maintenance

**Table 4 sensors-25-03110-t004:** Summary of Radio Mobile simulation scenarios.

Scenario	Brief Description	Distance (m)	Tx/Rx Height (m)	Environmental Characteristics
1	4° slope profile, LoS evaluation	336	2 / 1	Sloped terrain, no major obstructions
2	Real case with observed loss at 336 m	336	10 / 1	Moderate vegetation, link failure despite sufficient power
3	Communication across a ravine	370	10 / 1	Rx in lower areas, dense vegetation, vertical obstruction
4	Open area (control)	490	10 / 1	Clear terrain, good alignment with transmitter
5	Forested section	490	10 / 1	Dense vegetation, same profile as Scenario 4 for comparison
6	Extended path with vegetation	560	10 / 1	Similar elevation to Tx, light vegetation, most distant point

**Table 5 sensors-25-03110-t005:** Details of simulation scenarios.

S	Latitude (Rx)	Longitude (Rx)	TxH (m)	RxH (m)	D (m)	OD (m)	Path Loss (dB)	Rx Power (dBm)	Observation
1	−0.161931	−78.456219	2	1	336	90	102.6	−84.6	Simulated with identical Tx and Rx height.
2	−0.161931	−78.456219	10	1	336	320	91.8	−73.8	Real case, data loss observed at 91.58 dB.
3	−0.162470	−78.455889	10	1	370	90	106.2	−88.2	Receiver located deeper within the ecological trail near the ravine (Simulated).
4	−0.163669	−78.454818	10	1	490	480	88.7	−70.7	Receiver positioned above the trail, near Tx height, with vegetation (Simulated).
5	−0.163669	−78.454818	10	1	490	480	88.7	−70.7	Same location as scenario 4, without vegetation (Simulated).
6	−0.163611	−78.454160	10	1	560	570	93.3	−75.3	Farthest point, aligned with Tx height, with vegetation (Simulated).

Note: S—Scenario; TxH—Tx Height; RxH—Rx Height; D—Distance; OD—Obstruction Distance.

## Data Availability

The dataset is available upon request from the authors.

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
