# Peer review of "Implementation of a Remote Monitoring Station for Measuring UV Radiation Levels from Solarimeters Using LoRaWAN Technology"

_sensors, 2025, doi:10.3390/s25103110_

Round 1

Reviewer 1 Report

Comments and Suggestions for Authors

The overall quanlity of the paper is high and it has the merit to be of public interest. I have only one question, you said that the system was configured to operate from 8 AM to 6 PM and then enter deep sleep mode to conserve energy and comply with the ITU recommendations, but is this sufficient to comply with the 1% duty cycle regulation regarding LoRaWAN?

Author Response

Thanks to the reviewers for their comments and suggestions, which are addressed in the attached file.

Reviewer 2 Report

Comments and Suggestions for Authors

The article presents an interesting and up-to-date approach to monitoring ultraviolet (UV) radiation using LoRaWAN technology. The authors developed a prototype system that enables remote UV radiation measurements and transmits the data to the Ubidots cloud platform, thus enabling data visualization and analysis. The article provides well-documented use of this technology in a real university environment (Universidad de las Américas). Under test conditions, the system showed high effectiveness, with only a 3% deviation from the results of professional measuring systems. Comparison with data from the Environmental Secretariat showed a 97% correlation and a low relative error (2.67%), confirming that the system can serve as a cost-effective alternative to commercial solutions, especially in the context of health and environmental protection.

The authors should add the following information to the article:

Sensitivity characteristics, measurement range, accuracy, and power supply requirements of the system.

Calibration – whether and how the calibration was performed with reference to professional equipment. Average energy consumption of the measurement node in active and standby states. Type of power source used (battery, solar panel). Whether energy-saving strategies were implemented (e.g., ESP32 deep sleep mode, cyclic measurements, etc.). Justification for selecting specific locations (e.g., exposure to sunlight, availability of LoRa network coverage).

Author Response

(The authors gave the same response as above.)

Reviewer 3 Report

Comments and Suggestions for Authors

The paper focuses to the implementation of LoRaWAN technology for a specific application. The applications is interesting, but a good part of the paper is presenting the development of the LoRa hardware transmitter/receivers, something that has no significant scientific interest since this protocol, is already well established and analyzed.   There are numerous of the self solutions for the implementation of this technology, and therefore this paper does not add any novel scientific knowledge for this technology. In addition measurements conducted into the University for the range efficiency of the network is also limited to few hundreds meters, when it is widely known that LoRa networks can achieve  ranges up to 20 Km.

Therefore the paper, if it was to be accepted, should focus to the evaluation of the application itself by conducting many test trials in various settings and skip the part of the LoRa network development. In addition the SWOT analysis presented is not well substantiated  since the percentages appointed to each element seems arbitrary

Author Response

(The authors gave the same response as above.)

Round 2

Reviewer 3 Report

Comments and Suggestions for Authors

The paper meets the review comments and can be published.